# Intake of Natural, Unprocessed Tiger Nuts (*Cyperus esculentus* L.) Drink Significantly Favors Intestinal Beneficial Bacteria in a Short Period of Time

**DOI:** 10.3390/nu14091709

**Published:** 2022-04-20

**Authors:** Marta Selma-Royo, Izaskun García-Mantrana, M. Carmen Collado, Gaspar Perez-Martínez

**Affiliations:** Laboratory of Lactic Acid Bacteria and Probiotics, Department of Biotechnology, Institute of Agrochemistry and Food Technology (IATA-CSIC), Av. Agustin Escardino 7, 46980 Paterna, Valencia, Spain; mselma@iata.csic.es (M.S.-R.); igama@iata.csic.es (I.G.-M.); mcolam@iata.csic.es (M.C.C.)

**Keywords:** microbiota, tiger nut, resistant starch, polyphenols, Mediterranean diet

## Abstract

Horchata is a natural drink obtained from tiger nut tubers (*Cyperus esculentus* L.). It has a pleasant milky aspect and nutty flavor; some health benefits have been traditionally attributed to it. This study evaluated the effects of an unprocessed horchata drink on the gut microbiota of healthy adult volunteers (*n* = 31) who consumed 300 mL of natural, unprocessed horchata with no added sugar daily for 3 days. Although there were no apparent microbial profile changes induced by horchata consumption in the studied population, differences could be determined when volunteers were segmented by microbial clusters. Three distinctive enterogroups were identified previous to consuming horchata, respectively characterized by the relative abundances of *Blautia* and *Lachnospira* (B1), *Bacteroides* (B2) and *Ruminococcus* and *Bifidobacterium* (B3). After consuming horchata, samples of all volunteers were grouped into two clusters, one enriched in *Akkermansia*, *Christenellaceae* and *Clostridiales* (A1) and the other with a remarkable presence of *Faecalibacterium, Bifidobacterium* and *Lachnospira* (A2). Interestingly, the impact of horchata was dependent on the previous microbiome of each individual, and its effect yielded microbial profiles associated with butyrate production, which are typical of a Mediterranean or vegetable/fiber-rich diet and could be related to the presence of high amylose starch and polyphenols.

## 1. Introduction

The consumption of vegetable products has notably grown in recent years, and the commercial offering of soy, vegetable drinks and smoothies has grown accordingly. Tubers of *Cyperus esculentus*, also known as earth almond or tiger nuts, are used to make horchata, a pleasant milky drink obtained from ground, rehydrated tubers, that is very popular in Spain and other Mediterranean countries; seasonal visitors have internationally spread its appeal in the last decade. It is a traditional product in the land of Valencia (Spain), with a protected designation of origin (PDO), where it was introduced by the Arabs, likely before the 8th century [1]. In some countries, it is considered a very invasive weed [2], a fact that may explain the English name of “tiger nut”. This product has a specific volatile composition that explains its characteristic almond-like sensorial properties [3], and it is naturally sweet as it contains 9.2–13.0 g sucrose/100 mL of small, variable amounts of glucose and fructose, in addition to 2–3 g/100 mL of starch granules in suspension. Horchata also contains an emulsion of oil (2.4–3.1% *v/v*) with abundant phospholipids (0.5–0.6 g/100 mL) and a fatty acid composition similar to olive oil (11.6–22.3% saturated, 65.6–76.1% monosaturated, 9.2–13.6% polyunsaturated fatty acids) [4,5,6,7]. This tiger nut product also has a low amount of proteins and amino acids (0.6–1.4 g/100 mL protein), where L-arginine can reach 25% and 55% of total and free amino acids, respectively [6,7,8]. Furthermore, it contains vitamin C, biotin and vitamin E [7,9] as well as a range of polyphenols, such as gallic acid, 3,4-dihydroxybenzoic acid, catechin, rutin or coumaric acid, which have been detected in relatively high concentrations (1–40 mg/L), as well as a variety of conjugated polyphenols and other unidentified compounds [10,11,12]. However, as expected, the composition of horchata changes with the geographical origin of the tiger nut tubers [7,9] and also depends on the processing conditions as some components of horchata are sensitive to thermic treatment, demonstrated by high resolution mass spectrometry [9].

Gut microbiota and its functionality are recognized as crucial elements for maintaining human health [13]. Alterations in gut microbiota are commonly associated with several noncommunicable diseases, such as obesity [14], diabetes mellitus type 2 [15], atopy [16] and even neurological disorders [17]. Due to the great impact of the diet in shaping the gut microbiota composition [18], nutritional interventions have been proposed as one of the strategies that could contribute to the improvement of the treatment/prevention of these diseases through the microbiota modulation.

Horchata has traditionally shown diuretic, digestive and antidiarrheic properties [19]; however, there is no scientific evidence that relates its intake to disease prevention, any reported health marker or any potential activity as a microbiota modulator. The objective of this work was to determine changes in the microbiota after a 3-day intervention with a freshly prepared, natural (no enzymatic or thermic treatments, nor added sugars) horchata drink in order to pinpoint possible relationships between the benefits traditionally attributed to this drink.

## 2. Materials and Methods

### 2.1. Tiger Nut Drink Preparation

Commercial, natural horchata (nonpasteurized) was supplied by Món Orxata, S.L., and it was prepared according to the traditional method, without the addition of sugar. Briefly, curated tiger nut tubers (dried and stored for at least 4 months) were hydrated for at least 24 h, and then they were thoroughly washed with chlorinated water and rinsed. Immediately after, the tubers were milled with at least 3 volumes of water and stirred. The fiber residues were separated by filtration from the milky water extract. Additional water was added to adjust the fat content to about 4% (Appendix A).

### 2.2. Study Design and Participants

A total of 35 healthy adults (Valencia, Spain) participated in the study. The inclusion criteria was >18 years old, nondeclared gastrointestinal or immune-related pathology and zero consumption of antibiotics, medication or pre/probiotics during the previous 2 months before the study.

The intervention consisted of a daily intake of 300 mL of a natural, unprocessed drink of tiger nuts (*Cyperus esculentus* L.) for 3 days, consumed in the morning. At the beginning (B, before) and the end (A, after) of these time points, volunteers offered clinical, nutritional and anthropometric information as well as self-collected fecal samples following an informed protocol. Additionally, the intake of probiotics and prebiotics was restricted throughout the intervention study. The BMI was calculated, and volunteers were classified as lean or normal weight (≤25.0 kg/m^2^), overweight (25.0–30.0 kg/m^2^) or obese (≥30.0 kg/m^2^) [18]. Experiments were carried out following approved guidelines and regulations. All subjects gave their informed consent for inclusion before they participated in the study. The study was approved by the ethics committees from the Servicio Valenciano de Salud (Reference no.52327) (Spain), a substudy approved on 31 March 2016.

### 2.3. Nutritional Assessment

The dietary intake of the participants was determined through a validated, comprehensive 140-item food frequency questionnaire (FFQ) [19]. In all cases, we validated the FFQ information registered by participants with a 3-day food record questionnaire for the intake of dietary nutrients. The FFQ records were transformed to daily energy, macro- and micronutrients intakes using the nutrient food composition tables developed by the Centro de Enseñanza Superior de Nutrición Humana y Dietética (CESNID) (http://www.cesnid.ub.edu/es/index.htm, accessed on 18 October 2018) and analyzed by EASY DIET software (https://www.easydiet.es/, accessed on 1 September 2020).

Additionally, a 14-item, PREDIMED (PREvención con DIeta MEDiterránea, http://www.predimed.es/investigators-tools.html, accessed on 30 March 2016)-validated test was used to appraise the adherence of participants to the Mediterranean diet (MD) [20]. The MD score ranged from 0 (minimal adherence) to 14 (maximal adherence). A score of 9 or more points meant good adherence to the Mediterranean diet.

### 2.4. Biological Samples and Gut Microbiota Analysis

A fecal sample was collected by each volunteer at home using plastic containers and was frozen at −20 °C before sending to the laboratory, where the samples were stored at −80 °C until analysis.

The fecal sample (100 mg) was used for the total DNA extraction following a commercial kit, the Master-Pure DNA Extraction Kit (Epicentre, Madison, WI, USA), as described elsewhere [21]. The total DNA was purified using the DNA Purification Kit (Macherey-Nagel, Duren, Germany), and the total concentration was measured using a Qubit^®^ 2.0 Fluorometer (Life Technology, Carlsbad, CA, USA). The targeted 16S rRNA amplicon (V3–V4 region) was sequenced following Illumina protocols [21]. Briefly, a multiplexing step was conducted using the Nextera XT Index Kit (Illumina, San Diego, CA, USA), and the correct amplicon was checked with a Bioanalyzer DNA 1000 chip (Agilent Technologies, Santa Clara, CA, USA). Libraries were obtained (2 × 300 bp paired-end run, MiSeq Reagent kit v3) on a MiSeq-Illumina platform (FISABIO sequencing service, Valencia, Spain). The QIIME1 pipeline was used to process the raw sequences to operational taxonomic units [22]. Chimeric sequences and those not aligned were removed from the data set as were the sequences classified as cyanobacteria and chloroplasts. The raw sequences obtained are available in the NCBI BioProject database under accession number PRJNA816156.

### 2.5. Statistical Analysis

Clinical, nutritional and anthropometric data differences were tested with the *T*-test and the Mann−Whitney analysis according to data normality assessed by the Shapiro−Wilk test in Graphpad software v.5.04 (GraphPad Software, La Jolla, CA, USA). The chi-squared test (2 × 2) was performed to assess the differences in the categorical variables.

Microbiota clustering was generated at the genus level, as described elsewhere [23], using the phyloseq [24], cluster [25], MASS [26], clusterSim [27] and ade4 R packages [28]. Briefly, the Jensen−Shannon distance and partitioning around medoid (PAM) clustering were used. The optimal number of clusters was calculated with the Calinski−Harabasz (CH) index.

Alpha diversity indexes (Chao1 and Shannon indexes) were determined, and differences between the groups were assessed with the Kruskall−Wallis ANOVA test through the MicrobiomeAnalyst platform [20,29]. The beta diversity was based on the Bray−Curtis distance (nonphylogenetic), and permutational multivariate analysis of variance (PERMANOVA) was used to test the significance between the groups; the multivariate redundancy discriminant analysis (RDA) was also carried out. Data were considered statistically significant at *p*-value < 0.05, and comparisons were adjusted by the false discovery rate (FDR). The linear discriminant analysis effect size (LEfSe) was used to detect bacterial features that differed between clusters and intervention times [30].

## 3. Results

### 3.1. Association of Microbiota Composition with Diet and Anthropometric Data

From a total number of 35 healthy adults, 4 individuals abandoned the study due to intestinal discomfort and short intestinal transit (diarrhea). The remaining 31 healthy individuals were included in the study and reported no problems or changes in the intestinal habits. A total of 62 fecal samples were analyzed, which corresponded to fecal samples before and after 3 days of intervention. When all samples were analyzed, no significant impact was found in the gut microbial communities due to horchata intake, per the redundancy analysis (RDA), F = 0.41 and *p*-value > 0.05, and Bray−Curtis distances PCoA (Appendix A). Interestingly, before the intervention samples were grouped into three different clusters according to their microbial composition (Figure 1A) but after horchata intake, a cluster analysis of the microbiota yielded two groups (Figure 1B). This reduction of microbial clusters indicated a clear effect on the gut microbial composition, at least in a fraction of the volunteers.

No significant differences in clinical and anthropometric data were found between volunteers belonging to different microbial clusters (Table 1). Furthermore, the microbial clusters at baseline were not influenced by dietary intake, as assessed by FFQ data shown in Table 1. The FFQ was validated by a 3 d recall food record questionnaire for the intake of dietary nutrients (Appendix A). Drinking 300 mL of horchata during the morning coffee break represented an intake of 206.7 kcal (Appendix A), which was not considered an important change in the daily calorie intake.

### 3.2. Further Cluster Analysis of Microbiota before and after Horchata Intake

A redundancy discriminant analysis (RDA) on the genus level showed significant differences between the microbial populations of all different clusters (B1, B2, B3, A1, A2) (variance = 46.65, F = 1.52 and *p*-value = 0.001) (Figure 1C). This difference was confirmed by use of PERMANOVA based on the Bray–Curtis distance (model R2 = 0.136 and *p*-value = 0.0003). No differences in alpha diversity indexes were observed at baseline (Figure 1D,E). Three clusters of participants were identified at baseline according to microbiota composition (B1, *n* = 9; B2, *n* = 16; and B3, *n* = 6).

Interestingly, relative abundance heatmaps showed that cluster B1 was characterized by a significant abundance of two *Lachnospiraceae* genera, *Blautia* (13.85%) and an unclassified *Lachnospiraceae* (9.3%); cluster B2 had a higher presence of *Bacteroides* (9.31%), and cluster B3 had a higher presence of *Ruminococcus* (7.82%) and *Bifidobacterium* (11.32%) (Figure 2).

A similar analysis tracked the microbiome of samples after the intake of horchata for 3 days. As mentioned above, the gut microbial composition of volunteers changed after intervention, giving rise to two distinct groups (A1, *n* = 14 and A2, *n* = 17) (Figure 1B). The RDA showed that bacterial communities were significantly affected, with B3 as the most distinct cluster (F = 1.43 and *p*-value = 0.009) (Figure 1C). This difference was confirmed by use of PERMANOVA based on the Bray–Curtis distance (model R2 = 0.107 and *p*-value = 0.001). Cluster A1 had a higher microbial richness (Chao1 index, *p*-value = 0.011) and diversity (Shannon index, *p*-value = 0.092) than A2 (Figure 1D,E).

At the genus level, cluster A1 was characterized by a significantly higher presence of *Akkermansia*, *Oscillospira* and the unclassified genus of Christensenellaceae (Clostridiales) while these genera were reduced in cluster A2, which was characterized by the presence of *Faecalibacterium, Bifidobacterium, Collinsella, Lachnospira* and an unclassified *Peptostreptococcaceae* genus (Figure 2 and Appendix A).

When all clusters before and after intervention were compared, significant differences in microbial communities could be found (RDA F = 1.74 and *p*-value = 0.009, Figure 1C), and clusters were grouped according to the microbial composition, independent of the individual. That is, cluster B2 was grouped with cluster A2, and clusters B1 and B3 were grouped with cluster A1 (Figure 1E). In fact, there was a significant conversion of enterotype B3 (90% of individuals) to A2 and B1 (73% of individuals) to enterotype A1 while enterotype B2 (*Bacteroides*) migrated almost equally to A1 and A2 (Figure 3). Differences in the microbiota composition at the genus level were observed within the migration pairs (Figure 4A), but there were little differences in alpha diversity (Figure 4B,C).

## 4. Discussion

The consumption of *Cyperus esculentus* L. tubers (tiger nut) is widespread in the Mediterranean basin, Middle East and some African countries as a sweet snack, side dish or horchata, the drink obtained from ground, hydrated tubers. *C. esculentus* contains a great variety of antioxidants and polyunsaturated fatty acids; hence, some health properties have been attributed to this plant [7]. In fact, different countries use various species of the genus *Cyperus*, such as *Cyperus articulatus* [21,22,31] and *Cyperus rotundus* [31,32], in traditional medicine. Nevertheless, there is no evidence supporting such benefits for *C. esculentus*, and at present, it is consumed for its excellent nutritional and sensory qualities.

In the present work, specific microbial population changes have been found in healthy adults after a short intervention of 3 days, as reported in other short intervention studies of 2 or 3 days with a diet change from a meat-based to a vegetable-based diet [18] or with walnut intake [33]. Furthermore, relevant changes at the genus level can be observed in periods as short as 24–48 h in a dietary intervention [34]. In our study, despite no apparent overall changes in microbiota profiles associated with horchata intake in this study, three characteristic microbial clusters were found at baseline: B1 was enriched in the genera *Blautia* and *Lachnospira* (Lachnospiraceae), B2 in *Bacteroides* and B3 in *Ruminococcus* and *Bifidobacterium* (Figure 2). Those three patterns were distinctly modified by the intervention and migrated to two bacterial distinct profiles (A1, A2), dominated by the bacterial genera associated with a vegetable-rich diet or adhesion to a typical Mediterranean diet. A1 showed an increased presence of the bacterial genera *Akkermansia, Oscillospira* and *Christensenellaceae,* and, coincidently, an increase of *Akkermansia* and *Christensenellaceae* was associated with the adhesion to a Mediterranean diet [35]. Group A2 was enriched in *Faecalibacterium, Bifidobacterium, Collinsella, Lachnospira* and, with the exception of *Collinsella*, all other genera were associated with a healthy microbiome profile and butyrate production, typical of vegetable- and fiber-rich diets. In fact, research evidence associated vegetarian diets to the abundance of the genera *Roseburia, Lachnospira* and *Prevotella* as well as some species such as *Eubacterium rectale* and *Ruminococcus bromii*. High adherence to a Mediterranean diet also led to a remarkable presence of *Lachnospira* and *Prevotella*. In both diets, there was a significantly higher content of gut butyrate, propionate and acetate, compared to the content in individuals who followed omnivorous diets, where *Ruminococcus* (also of the *Lachnospiraceae* family) dominates [18,36,37].

The potential explanation for the observed changes could be bound to the high content of insoluble starch granules in horchata of relatively small size (6–11 µm) with a moderate to high proportion of amylose (up to 19–21%) as well as their compact structure [38,39]. Those are the characteristics of resistant starch (RS), possibly class RS2 [23], but the functional properties of amylose-rich starch may be due to other factors, such as the ability to form complexes with lipids or proteins [24,38]. RS is considered a prebiotic with consolidated evidence on the beneficial effects on health [25,26], and its ability to modify the composition of gut microbiota has been known for a long time [23], with an impact on microbiota that is different to other complex polysaccharides [27]. RS promoted the gut colonization of *Akkermansia*, *Lactobacillus*, *Bifidobacterium*, Clostridial clusters IVg and XIVa+b, in rats [28], *Bacteroidetes, Bifidobacterium, Akkermansia* and *Allobaculum*, in mice [40] or *Prevotellaceae* and *Faecalibacterium*, in vitro fermentations [41]. The role of primary degraders of RS has been attributed to *Ruminococcus bromii* and *Bifidobacterium adolescentis* [42]. Nevertheless, when tested in humans, those two species increased in fecal samples under a RS2 diet but only if they were previously present at baseline, which occurred with butyrate-producing bacteria [43]. This stressed that RS2-rich diets can change the community structure within the subjects, which conditions the effects of RS2 to the previous microbiota composition. Finally, polyphenols in horchata may also impact gut microbiota composition. Gut microbiota service living organisms at the nutritional level by the synthesis of essential compounds, and its implication in the modification of compounds has consequent physiological effects, as in the case of the activation of polyphenols and plant-derived isoflavones [44]. Soy-derived foods and drinks have a longstanding tradition in oriental countries, and they have become a reference of health-promoting products, globally generating a drag effect on the consumption of other vegetable products [45,46]. Diets containing soy products increased the levels of beneficial bacteria, such as *Bifidobacterium*, *Lactobacillus* and *Faecalibacterium prausnitzii*, and reduced the overall ratio of Firmicutes/Bacteroidetes, indicators of a healthy microbiome [47]. Polyphenol-rich vegetable products may induce changes in the microbiome due to their antimicrobial activity. Vegetable polyphenols can be found as homo- or heteropolymers or glycosylated, and they can interact with gut microbiota and be modified by a number of microbial enzymes, such as different hydrolases, isomerases, dehydroxylases, decarboxylases, demethylases, etc. [48,49,50,51]. A polyphenol analysis of *C. esculentus* extracts indicated a high content of gallic acid, 3,4-dihydroxybenzoic acid, catechin and, to a lesser extent, rutin and coumaric acid [11,12]. These polyphenol concentrations may contribute to the alterations in the gut microbiota described in our microbiota analysis as green tea catechin, epigallo-catechin-3-gallate (EGCG) impedes the growth of both Gram-positive and Gram-negative bacteria by interfering with the membrane structure and by inhibiting oxidoreductases in the respiratory chain that are required for DNA synthesis [52,53,54,55,56,57].

Our results highlight that intraindividual variability of the gut microbiota composition could be a determinant for the outcome of a dietary intervention. Thus, the description of the basal microbiota composition is essential in any intervention study aiming to assess the impact of diet or other factors in the modulation of gut microbiota. This reinforces the evidence of interventions (dietary, clinical, etc.) on responders and nonresponders depending on the microbiota, opening new possibilities to advance personalized nutrition and medicine in future studies. On these grounds, it can be stressed that the intake of traditional and locally produced foods, such as horchata, significantly contributes to microbial profiles associated with geographical communities, specifically to the Mediterranean-diet-bound microbiota. The evidence described above underlines important aspects that should be afforded in future research. It would be very interesting to determine the role of specific taxa in the consistency and resilience of the microbiota during interventions, to dissect the role of resistant starch and polyphenols from dietary components and to focus on the long-term effect of this and other highly consumed local products on the microbiome constitution and correlated health effects.

## 5. Conclusions

This study describes a fast change (3 days) in the gut microbiota composition just by the intake of one dose per day of a tiger nut vegetable drink. Changes observed in the bacterial populations indicate a migration of all microbiome clusters towards microbial patterns similar to Mediterranean/vegetarian groups, and following the profile described after RS2 interventions, they very much depended upon the resident microbiota composition at baseline. Hence, these changes could be induced by the rich composition of starch powder in horchata with a likely contribution of native tiger nut polyphenols. Our results open the door to new interventional studies with added value to traditional products, such as “natural horchata”, with the potential to modulate the gut microbiota. Furthermore, the study highlights the basal microbiota composition as a determinant factor of the impact of a nutritional intervention in the gut microbiota.

## Figures and Tables

**Figure 1 nutrients-14-01709-f001:**
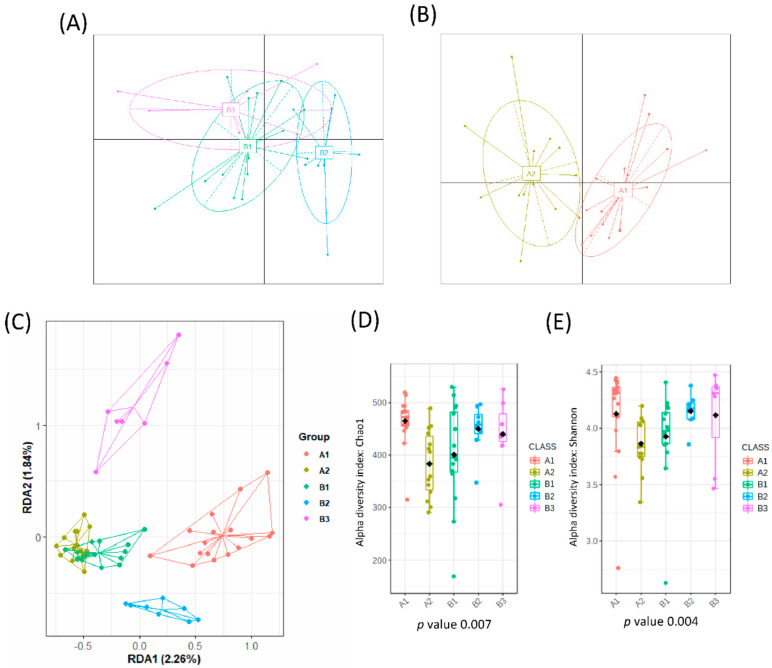
Microbial clusters and representative microbial taxa at the genus level. The partitioning around medoid method showed that volunteers were clustered at baseline into 3 groups (**A**) and into 2 groups after intervention (**B**) as reported in the Principal Coordinate Analysis (PCoA). (**C**) Multivariate redundancy discriminant analysis (RDA) showed distinct microbial communities for each group with the alpha diversity indexes: (**D**) richness (Chao1 index) and (**E**) diversity (Shannon index) depending on each cluster. Schemes follow the same formatting.

**Figure 2 nutrients-14-01709-f002:**
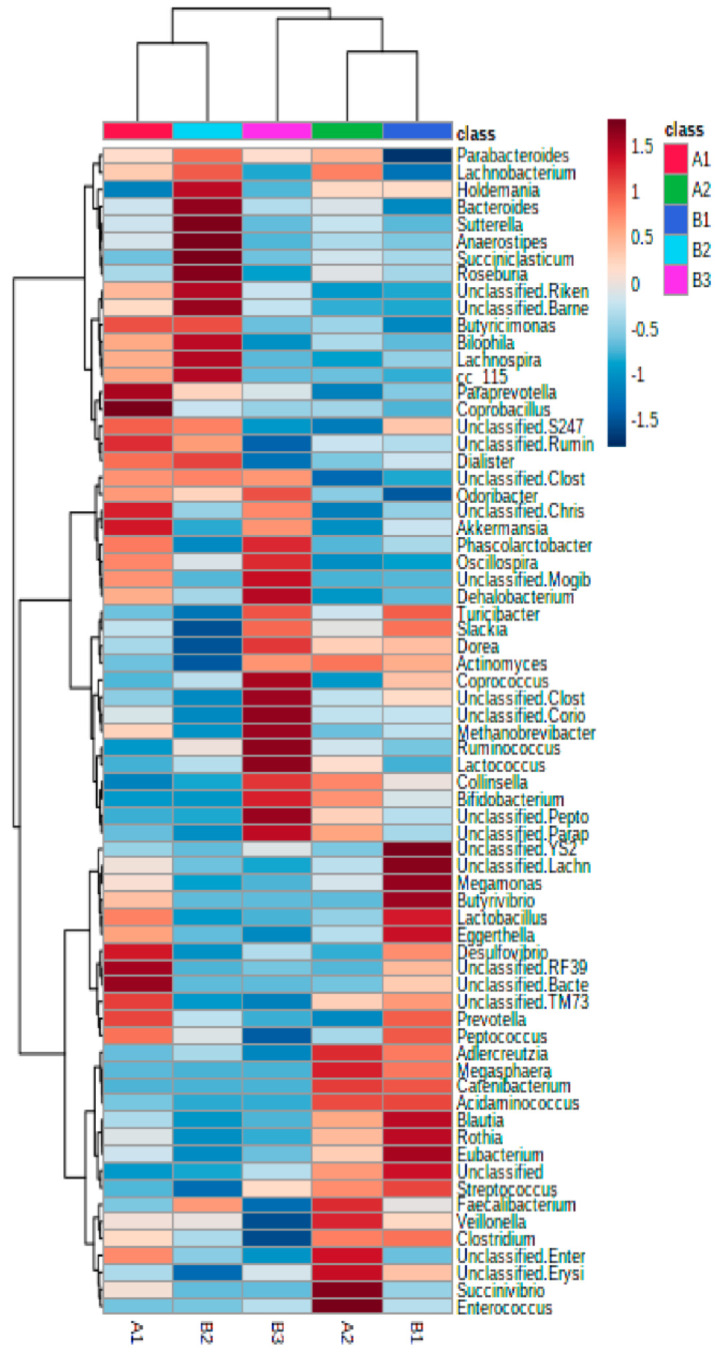
Hierarchical clustering heatmap showing the strongest contributors (Euclidean distance and Ward clustering algorithm).

**Figure 3 nutrients-14-01709-f003:**
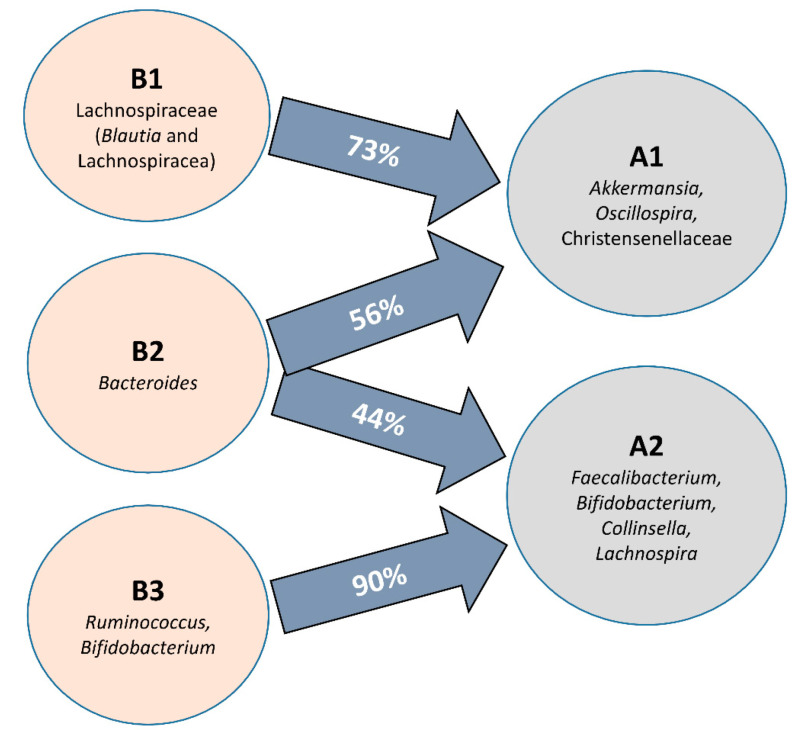
Diagram showing the proportion of participants whose microbiota composition migrates from the baseline groups (B1–3) during the intervention to new microbial profiles after horchata consumption (A1–2) at the genus level.

**Figure 4 nutrients-14-01709-f004:**
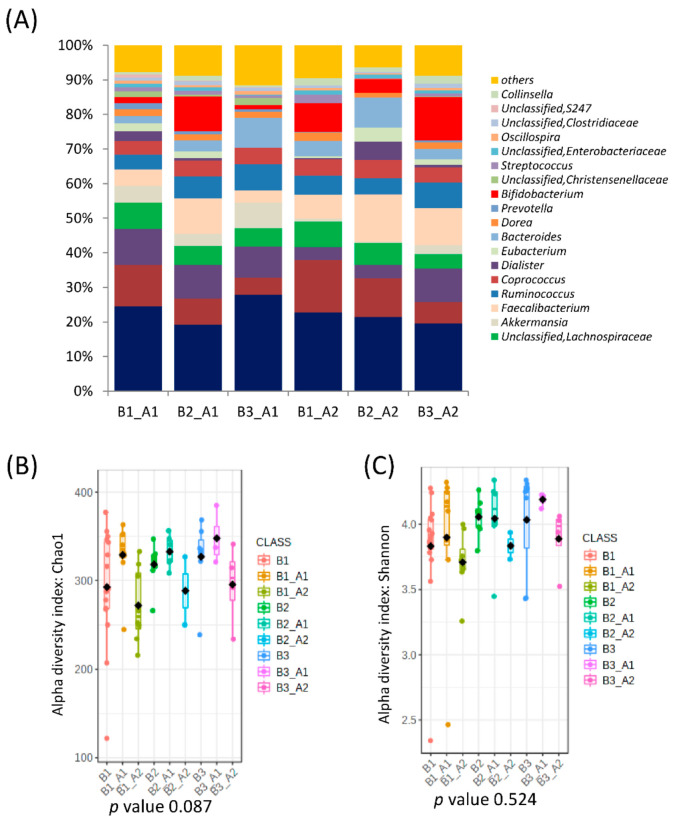
Microbial clusters and representative microbial taxa at the genus level according to migration groups between baseline clusters and after the intervention. (**A**) Relative abundances at the genus level according to migration groups between baseline clusters and after the intervention. (**B**,**C**) Microbial αdiversity indexes according to microbial cluster change after intervention richness (Chao1 index) (**B**) and diversity (Shannon index) (**C**).

**Table 1 nutrients-14-01709-t001:** Characteristics and specific dietary intake of the healthy volunteers involved in the 3-day intervention study.

	All (*n* = 31)	Cluster B1 (*n* = 16)	Cluster B2 (*n* = 8)	Cluster B3 (*n* = 7)	*p*-Value
Clinical Characteristics					
Age (years)	37.9 ± 11.2	35.6 ± 12.7	39.0 ± 8.9	42.0 ± 10.0	0.451
Gender (Female %)	15 (48.39)	8 (50.02)	8 (37.50)	4 (57.14)	0.737
BMI (kg/m^2^)	23.2 ± 3.3	23.2 ± 3.7	24.4 ± 2.8	21.7 ± 2.3	0.294
MD score	8.6 ± 1.8	8.6 ± 2.0	8.5 ± 2.1	8.8 ± 1.2	0.925
Dietary intakes					
Energy (kcal)	2959.5 ± 623.1	2907.2 ± 473.3	2801.1 ± 599.8	3260.1 ± 902.8	0.334
Total protein (g/day)	116.9 ± 27.6	112.1 ± 20.1	111.3 ± 24.7	134.4 ± 40.2	0.164
Animal protein (g/day)	74.4 ± 21.7	72.6 ± 17.8	69.4 ± 19.4	84.3 ± 31.3	0.386
Vegetal protein (g/day)	42.5 ± 14.9	39.5 ± 10.5	41.8 ± 14.2	50.1 ± 22.7	0.300
Lipids (g/day)	133.1 ± 27.8	134.8 ± 24.9	126.1 ± 21.9	137.4 ± 40.8	0.706
Cholesterol (g/day)	389.0 ± 109.5	386.8 ± 78.1	366.3 ± 112.7	420.0 ± 167.7	0.649
SFA	36.3 ± 10.2	37.2 ± 9.9	33.2 ± 8.7	37.6 ± 13.1	0.633
MUFA	64.4 ± 12.9	64.9 ± 13.2	62.5 ± 11.4	65.7 ± 15.7	0.879
PUFA	22.2 ± 9.0	22.5 ± 9.0	20.9 ± 7.7	23.1 ± 11.4	0.889
Carbohydrates (g/day)	308.5 ± 95.5	298.6 ± 82.1	288.6 ± 93.8	353.9 ± 123.9	0.362
Polysaccharides (g/day)	172.4 ± 75.7	166.2 ± 65.8	166.8 ± 78.2	193.1 ± 100.5	0.728
Total dietary fiber (g/day)	33.1 ± 12.1	31.6 ± 7.9	31.5 ± 10.9	38.6 ± 19.7	0.414
Vitamin A (mcg/day)	1243.3 ± 520.8	1312.5 ± 507.9	1024.6 ± 518.9	1335.2 ± 557.7	0.398
Retinoids (mcg/day)	325.5 ± 132.7	341.4 ± 128.6	286.4 ± 117.9	333.1 ± 166.4	0.635
Carotenoids (mcg/day)	5499.37 ± 3018.4	5818.7 ± 3220.7	4422.0 ± 3135.2	6000.77 ± 2458.3	0.514
Vitamin D (mcg/day)	3.3 ± 1.5	3.6 ± 1.9	3.2 ± 1.1	2.7 ± 1.0	0.406
Vitamin E (mg/day)	18.7 ± 6.1	19.0 ± 6.5	17.6 ± 4.5	19.3 ± 7.3	0.845
Thiamine (mg/day)	2.0 ± 0.8	1.9 ± 0.5	2.0 ± 0.7	2.4 ± 1.3	0.408
Riboflavin (mg/day)	2.1 ± 0.4	2.1 ± 0.5	1.9 ± 0.3	2.3 ± 0.5	0.224
Niacin (mg/day)	27.8 ± 7.2	27.4 ± 5.5	26.2 ± 6.8	30.98 ± 10.8	0.430
Vitamin B6 (mg/day)	2.8 ± 0.8	2.8 ± 0.7	2.8 ± 0.9	3.11 ± 1.2	0.766
Folic acid (mcg/day)	539.3 ± 197.1	532.2 ± 174.6	505.1 ± 185.8	594.9 ± 269.0	0.679
Vitamin B12 (mcg/day)	8.6 ± 3.3	8.3 ± 3.0	7.7 ± 2.0	10.5 ± 4.7	0.228
Vitamin C (mg/day)	270.3 ± 146.0	240.1 ± 104.1	256.0 ± 127.1	355.6 ± 222.4	0.212

Numerical data expressed as mean ± SD. The Student’s *t*-test was performed to assess the significance of the differences in the dietary intakes between the cluster sections in numerical variables and categorical variables, respectively. *p* < 0.05 was considered statistically significant. MD: Mediterranean diet adherences value.

## Data Availability

Data shared are in accordance with consent provided by participants, and no confidential data have been compromised. All sequencing data sets are available in the NCBI BioProject database under accession number PRJNA816156.

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
