# Peer review of "Intake of Natural, Unprocessed Tiger Nuts (Cyperus esculentus L.) Drink Significantly Favors Intestinal Beneficial Bacteria in a Short Period of Time"

_nutrients, 2022, doi:10.3390/nu14091709_

Round 1
Reviewer 1 Report
The work presents an interesting possible application of a drink typical of the Mediterranean culture, which could be a good starting point for subsequent works.
However, there are several points to review:
- It should be explained exactly how the drink is obtained from the tuber, possibly with a flow-sheet
- The mechanism of action is very hypothetical, it would first of all be verified if resistant starch is present and in what quantity, the hypothesized one is a bit small to justify the effect; as well as polyphenols should be analyzed specifically in terms of mechanism of action
- Stool collection should also be characterized in relation to the Bristol scale, the way of producing feces can influence the final result
- The proposed intake of the drink is interesting, it would be necessary to evaluate the glycemic index and consider the caloric intake which is not negligible for 330ml
- The duration of the study is too short and in addition, it does not include a control/placebo, in 3 days there could also have been a fluctuation that does not depend on the product also because very large variances are noted.
Author Response
- It should be explained exactly how the drink is obtained from the tuber, possibly with a flow-sheet
Author´s answer: As suggested by the reviewer, brief description of the preparation of horchata from tiger nuts used by traditional producers, like the supplier Món Orxata, S.L., has been included in the first point of Material and Methods (L. 82-89).
- The mechanism of action is very hypothetical, it would first of all be verified if resistant starch is present and in what quantity, the hypothesized one is a bit small to justify the effect; as well as polyphenols should be analyzed specifically in terms of mechanism of action
Author´s answer: This is a very important issue. Some health benefits of the consumption of horchata have been inferred from the high content in polyunsaturated fatty acids and polyphenols but, as mentioned in the introduction, there are no supporting scientific evidences. This is the first work describing that the consumption of horchata has a moderate effect on the composition of the gut microbiota, which was really interesting, but we have tried to distance ourselves from speculative arguments. Certainly, a careful analysis of the content in polyphenols and resistant starch would allow some connection to the observed changes in microbiota; however, a definite cause-effect relationship will require an investigation where the different components would be tested separately. We hope this will be afforded in the near future.
- Stool collection should also be characterized in relation to the Bristol scale, the way of producing feces can influence the final result
Author´s answer: Thank you for your comment. Unfortunately, we did not collect the information related to Bristol scale before and after the intervention, but we selected volunteers with normal bowel habits and, through the intervention time, they had no alterations in the transit time, at least in those participants that remained in the study. Just four participants had intestinal problems, including diarrhoea with unknown causes, and abandoned the study. Following the reviewer´s comments, we have included this information in the paper (L. 165-167).
- The proposed intake of the drink is interesting, it would be necessary to evaluate the glycemic index and consider the caloric intake which is not negligible for 330ml
Author´s answer: This comment raises an interesting but unknown issue, since most commercial horchatas have a high content in free sugars due to the addition of sucrose and/or to the presence of dextrose from starch hydrolysis in some commercial products. New supplementary data have been added showing the nutrient composition of standard freshly prepared natural horchata without added sugars and with intact starch -as it was not treated with amylases-, as well as its calorie content (new Table S2). Hence, 300 ml portions of horchata (not 330 ml, now corrected) were supplied during the morning’s coffee brake through the intervention, which would represent an intake of 206.7 kcal per dose. We expected that this calorie intake, mostly in the form of fatty acids and starch, would not influence significantly the glycemic index. An appropriate sentence has been added (L. 182-184)
- The duration of the study is too short and in addition, it does not include a control/placebo, in 3 days there could also have been a fluctuation that does not depend on the product also because very large variances are noted.
Author´s answer: The duration of the study is indeed short, and this is one of the interesting features of it. Previous studies demonstrated that just two or three days of dietary intervention (David et al., Nature, 2014; Garcia-Mantrana et al., Nutrients, 2019; both mentioned in the manuscript) could bring in remarkable changes in the gut microbiota composition. Regarding the need for a randomized placebo control trial, we would like to stress that the objective of this study was the comparison of the microbiome before and after intervention where, favoured by the short duration of the study, each individual was its own control. Further, it could be anticipated that similar results could have been obtained in a study with a control/placebo group, if samples had also been taken before and after the intervention. In fact, one of our main conclusions was that in short dietary interventions the baseline information on the microbiota is essential. Of note, as highlighted in the main body of the text, this is pilot study with a limited sample size designed to identify the potential beneficial effect of horchata on healthy adult microbiota. Our work has shown that there are solid grounds to program further studies that may focus on health issues and specific components of horchata that will require control/placebo studies.
Reviewer 2 Report
The study investigated the intake of natural unprocessed tiger nuts (cyperus escu- lentus L.) drink significantly favors intestinal beneficial bacteria in a short period of time. This study evaluated the effects of unprocessed horchata drink on the gut microbiota of healthy adult volunteers. However, there are also several problems need major revision:
1. In the "introduction" section of the article, the authors need to give a more detailed introduction to the current status of the research on intestinal flora, and from the abstract of the article, we can see that the focus of this article is on the regulation of intestinal flora, and the authors should emphasize the importance of intestinal flora to human health to highlight the value of this study.
2. In the "Materials and methods" section of the article, the author needs to organize this section appropriately, such as adding or removing subheadings to make the article look more organized and not too confusing to read.
3. Figure 4. Microbial clusters and representative microbial taxa at genus level according to migration groups between baseline clusters and after the intervention. In this part, the author may add more in-depth discussion and compare their results with the manuscripts recently published in authoritative journals. The author also needs to make adjustments to the layout of the article images.
4. In the "Discussion" section of the article, the authors mention changes in intestinal flora in response to specific interventions, which is novel, but the authors need to discuss it in more depth, give more of their own views, and give an outlook for the future.
5. Authors are requested to carefully check the format of the references used in the article to ensure that the references are in the required format.
Author Response
- In the "introduction" section of the article, the authors need to give a more detailed introduction to the current status of the research on intestinal flora, and from the abstract of the article, we can see that the focus of this article is on the regulation of intestinal flora, and the authors should emphasize the importance of intestinal flora to human health to highlight the value of this study.
Author´s answer: Within the space limitations, we have included some sentences describing the relevance of “microbiota-food” interactions area and we have emphasized the relevance of the balanced microbiota for human health (L 62-66).
- In the "Materials and methods" section of the article, the author needs to organize this section appropriately, such as adding or removing subheadings to make the article look more organized and not too confusing to read.
Author´s answer: Thank you for your comment. We have modified the material and method section, including appropriate subheadings to facilitate the reading.
- Figure 4. Microbial clusters and representative microbial taxa at genus level according to migration groups between baseline clusters and after the intervention. In this part, the author may add more in-depth discussion and compare their results with the manuscripts recently published in authoritative journals. The author also needs to make adjustments to the layout of the article images.
Author´s answer:
Thank you for your comment. Certainly, we found differences in microbiota composition at genus level between the migration pairs and, as such, we mentioned it but we did not extend on the lack of diversity between groups. Many top quality works are published these days in authoritative journals that use similar methods. We honestly apologize, as we could not decrypt the message of the reviewer.
The layout and quality of figures has been notably improved.
- In the "Discussion" section of the article, the authors mention changes in intestinal flora in response to specific interventions, which is novel, but the authors need to discuss it in more depth, give more of their own views, and give an outlook for the future.
Author´s answer: As suggested by the reviewer, we have included more details on the relevance of dietary intervention targeting human microbiota composition as well as the potential future perspectives.
- Authors are requested to carefully check the format of the references used in the article to ensure that the references are in the required format.
Author´s answer: We have been checked the format of the references and now, they are in the requested format.
Round 2
Reviewer 1 Report
Th authors try to justify but they didn't make any change so my opinion is the same